# Helminth infection driven gastrointestinal hypermotility is independent of eosinophils and mediated by alterations in smooth muscle instead of enteric neurons

Haozhe Wang[1], Kristian Barry[1], Aidil Zaini[1], Gillian Coakley[1], Mati Moyat[1], Carmel P. Daunt[1], Lakshanie C. Wickramasinghe[1], Rossana Azzoni[1], Roxanne Chatzis[1], Bibek Yumnam[2], Mali Camberis[2], Graham Le Gros[2], Olaf Perdijk[1], Jaime P. P. Foong[3], Joel C. Bornstein[3], Benjamin J. Marsland[1], Nicola L. Harris[1]*

1 Department of Immunology, School of Translational Medicine, Monash University, The Alfred Centre, Melbourne, Victoria, Australia, 2 The Malaghan Institute of Medical Research, Victoria University, Wellington, New Zealand, 3 Department of Anatomy and Physiology, University of Melbourne, Melbourne, Victoria, Australia

* Nicola.Harris@monash.edu

**Data Availability Statement:** All relevant data are within the manuscript and its Supporting Information files.

## Abstract

Intestinal helminth infection triggers a type 2 immune response that promotes a 'weep-and-sweep' response characterised by increased mucus secretion and intestinal hypermotility, which function to dislodge the worm from its intestinal habitat. Recent studies have discovered that several other pathogens cause intestinal dysmotility through major alterations to the immune and enteric nervous systems (ENS), and their interactions, within the gastrointestinal tract. However, the involvement of these systems has not been investigated for helminth infections. Eosinophils represent a key cell type recruited by the type 2 immune response and alter intestinal motility under steady-state conditions. Our study aimed to investigate whether altered intestinal motility driven by the murine hookworm, *Nippostrongylus brasiliensis*, infection involves eosinophils and how the ENS and smooth muscles of the gut are impacted. Eosinophil deficiency did not influence helminth-induced intestinal hypermotility and hypermotility did not involve gross structural or functional changes to the ENS. Hypermotility was instead associated with a dramatic increase in smooth muscle thickness and contractility, an observation that extended to another rodent nematode, *Heligmosomoides polygyrus*. In summary our data indicate that, in contrast to other pathogens, helminth-induced intestinal hypermotility is driven by largely by myogenic, rather than neurogenic, alterations with such changes occurring independently of eosinophils. (<300 words)

## Author summary

Intestinal helminth infection is a global threat to those living in poverty without adequate sanitation. Expulsion of intestinal worms is driven by a host type 2 immune response,

**Funding:** N.L.H received a salary from the National Health and Medical Research Council (NHMRC) of Australia, no. SRF-B 1140313. H. W. was financially supported by a Research Training Scholarship from Monash University. This work was additionally supported by a Discovery Project grant from the Australian Research Council (ARC), no DP210101500 awarded to N.L.H. The funders had no role in the study design, data collection and analysis, decision to publish, or preparation of the manuscript.

**Competing interests:** The authors have declared that no competing interests exist.

characterised by increased eosinophils, that results in the intestinal hypermotility and mucus secretion that dislodge the worm from its luminal habitat. Intestinal motility is largely controlled by the local enteric nervous system (ENS) and can be regulated by close interactions between neurons and intestinal immune cells. Using *Nippostrongylus brasiliensis* as a model of murine hookworm infection, we investigated the contribution of the ENS and eosinophils to intestinal hypermotility and worm expulsion. Despite the critical role of the ENS in regulating typical intestinal function, very little alteration to ENS structure or function was observed following Nb infection. Instead, Nb infected animals displayed dramatically increased smooth muscle thickness and contractile strength. Alterations of smooth muscle were also observed in response to infection with *Heligmosomoides polygyrus*. Neither *Nippostrongylus brasiliensis*-induced intestinal hypermotility nor altered smooth muscle morphology required eosinophils. Our findings reveal that, in contrast to other intestinal pathogens, myogenic rather than neurogenic alterations drive small intestinal hypermotility and pathogen expulsion following intestinal helminth infection. (<200 words)

## Introduction

Hookworms are highly successful parasites infecting approximately 500 million people living in impoverished conditions causing anaemia and protein malabsorption [1]. The murine hookworm *Nippostrongylus brasiliensis* (Nb) elicits a strong type 2 immune response, characterised by release of type 2 cytokines, such as IL-4 and IL-13, and the recruitment or expansion of type 2 immune cells [2,3]. Type 2 immune responses promote: i) resistance against re-infection, ii) the repair of damaged tissues and iii) the expulsion of lumen dwelling worms through the 'weep and sweep' response [2]. The 'weep and sweep' response is particularly pronounced following Nb infection with mice expelling adult worms from the intestinal lumen within a week of the first infection [4]. This response is primarily driven by IL-14 and IL-13 and involves increased intestinal motility and mucous secretion [5]. In contrast to Nb, primary infection with the rodent nematode *Heligmosomoides polygyrus* (Hp), results in a chronic infection with animals unable to elicit a 'weep and sweep' response strong enough to dislodge worms when they first mature into lumen dwelling adults [6,7]. By contrast animals elicit a strong type 2 immune response following secondary infection with Hp leading to larval trapping and rapid expulsion of adult worms [6,7].

 The intestine is home to an intrinsic enteric nervous system (ENS) that controls motor functions, blood flow and epithelial absorption or secretion [8,9]. Neuronal control of motor functions involves complex motor activities that regulates the mixing and propulsion of luminal contents [8,9]. Different functional enteric neurons are grouped into two interconnected plexuses known as the myenteric and submucosal plexus. The myenteric plexus is located between the two layers of smooth muscle components of the intestine, known as the circular and longitudinal muscles [8,9]. The myenteric neurons are responsible for generating contractile patterns that directly influence intestinal motility and transit. The submucosal plexus can be found next to the lamina propria, and its main function is to regulate fluid secretion, absorption, and vasodilation [8,9]. It is well documented that pathogenic infections can alter the structure and activities of the ENS [10–16], yet very few studies have addressed these parameters during helminth infection despite evidence indicating that infection can alter the availability of neurotransmitters and neuropeptides within the intestine [17,18].

Eosinophils are one of the most abundant type 2 immune cells expanded in response to helminth infection [19,20]. They have been documented to trap and kill the larval forms of select helminths, including *Schistosoma mansoni* [21] and *Ascaris suum* [22], yet appear to be redundant for protective immunity against other helminths including Nb [19, 20]. By contrast, the presence of eosinophils promotes the survival of muscle larvae following primary infection of mice with *Trichinella spiralis* [23,24], further highlighting the complex roles these cells play in the host response to helminths. A growing number of studies present evidence for eosinophils as regulators of organ development, mucosal immune responses, and tissue regeneration [25]. A recent study from our laboratory demonstrated that intestinal eosinophils play a critical role in maintaining normal villous structure, nutrient uptake and intestinal motility in response to microbial colonisation [26]. Eosinophil deficiency also resulted in the downregulation of neuron-related genes in proximal jejunum and eosinophils formed intimate contacts with neuronal axons in jejunal villi, suggesting neuro-immune interactions may underpin at least some of the functions of intestinal eosinophils [26].

In this study, we tested the hypothesis that eosinophils function to regulate intestinal damage and/or hypermotility following hookworm infection using wildtype BALB/c and eosinophil deficient dblGATA1 mice [27]. Our findings instead show that eosinophils do not alter worm expulsion, villous atrophy or intestinal hypermotility following Nb infection. However, Nb infection was only associated with minor changes in ENS structure or function indicating that alterations to the ENS are not a major contributor to the intestinal hypermotility that occurs response to infection. Worm expulsion occurred alongside increased smooth muscle thickness and contraction strength indicating myogenic rather than neurogenic changes drive intestinal hypermotility. The finding of stronger myogenic than neurogenic alterations in response to helminth infection were also extended to another intestinal helminth, Hp.

## Results

### *N. brasiliensis* infection alters intestinal physiology in an eosinophil-independent manner

To determine the impact of Nb infection on intestinal physiology we examined a range of parameters including gross changes in small intestinal length, weight, and motility. Studies were performed in eosinophil-sufficient BALB/c, and eosinophil-deficient dblGATA1 mice. Infection increased both the length and weight of the small intestine at days 5 or 9 post-infection (p.i.) then returned to normal levels between day 28 and day 40 p.i (Fig 1A and 1B). Eosinophils did not contribute to Nb-induced alterations to the length or weight of the small intestine (Fig 1A and 1B).

In keeping with a proposed role for intestinal hypermotility in worm expulsion [2,3] we observed an approximately two-fold increase in the distance travelled by bolus of carmine red dye, delivered by oral gavage, at day 5 p.i. infection (Fig 1C). This timing coincided with the expulsion adult worms which occurred between day 5 and day 7 p.i. (Fig 1D). Small intestinal hypermotility remained high at day 9 p.i. and slowly returned to normal levels by day 40 p.i (Fig 1C). Nb induced intestinal hypermotility and worm expulsion were equivalent in BALB/c and dblGATA1 as was worm expulsion (Fig 1C and 1D), indicating eosinophils do not contribute to intestinal hypermotility. Of note small intestinal motility was not impacted by eosinophil deficiency even in naïve mice (Fig 1C), suggesting our previous observations of increased total gut transit times in dblGATA mice [26] is likely due to altered motility in the large intestine. In summary our data support a contribution of intestinal hypermotility to luminal worm expulsion and indicate that eosinophils are redundant for this process.

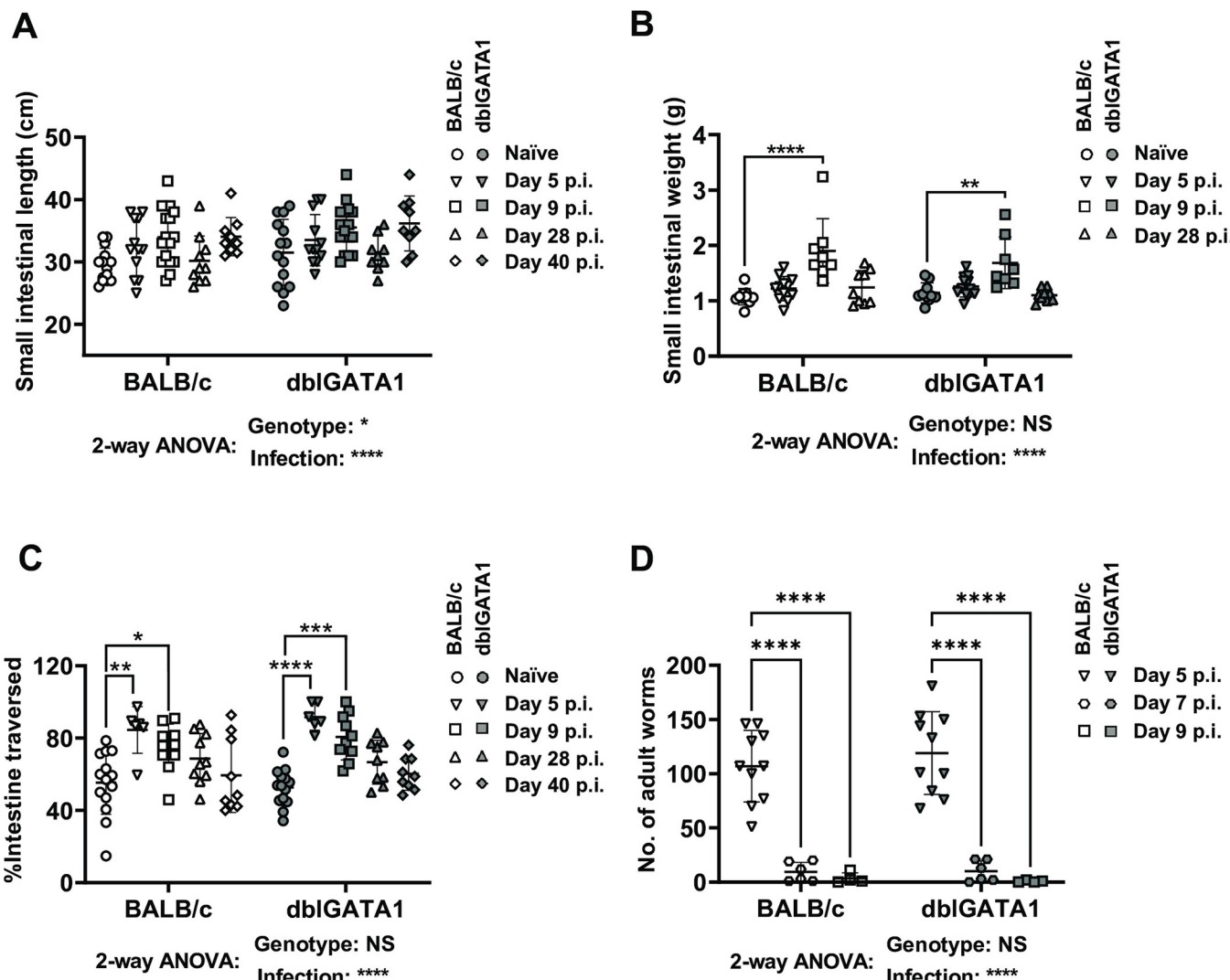

**Fig 1. Impact of Nb infection on small intestinal morphology and motility in BALB/c and dblGATA1 mice.** dblGATA1 mice and wildtype BALB/c controls were infected with 400–500 L3 Nb and sacrificed at the indicated days p.i. Control groups of naïve mice were included. A) Weight and (B) length of the entire small intestine were determined. (C) Intestinal motility was determined using a carmine red dye transit assay. Dye was administered by oral gavage 30 mins prior to sacrifice and motility determined as the distance travelled by dye (as a percentage of total small intestine length) at the time of sacrifice. (D) Number of luminal adult Nb worms was determined using a Baermann assay. For all data, symbols represent individual animals. Data in A-C are pooled from 2–3 independent experiments with n = 6–16 per timepoint for infected groups. Data in D are from n = 4–10 animals per timepoint and are pooled from 2 independent experiments. All data are shown as mean ± SEM and significance determined using a two-way ANOVA with Tukey's post-hoc analysis.

### *N. brasiliensis* infection results in villous atrophy in both eosinophil-sufficient and eosinophil-deficient mice

Based on the knowledge that hookworms cause protein malabsorption and damage the intestinal epithelium [1,28] we investigated whether mice infected with Nb exhibited villous atrophy using a previously established approach [26,29] of whole-mount microscope imaging that allows three-dimensional assessment of the villous architecture. Images are then utilised to determine villous atrophy, which is assessed by measuring the villous length and surface area (calculated from the vascular cage). Nb infection resulted in a transient increase in jejunal villous length and surface area at day 5 p.i. (Fig 2A–2C), followed by a dramatic

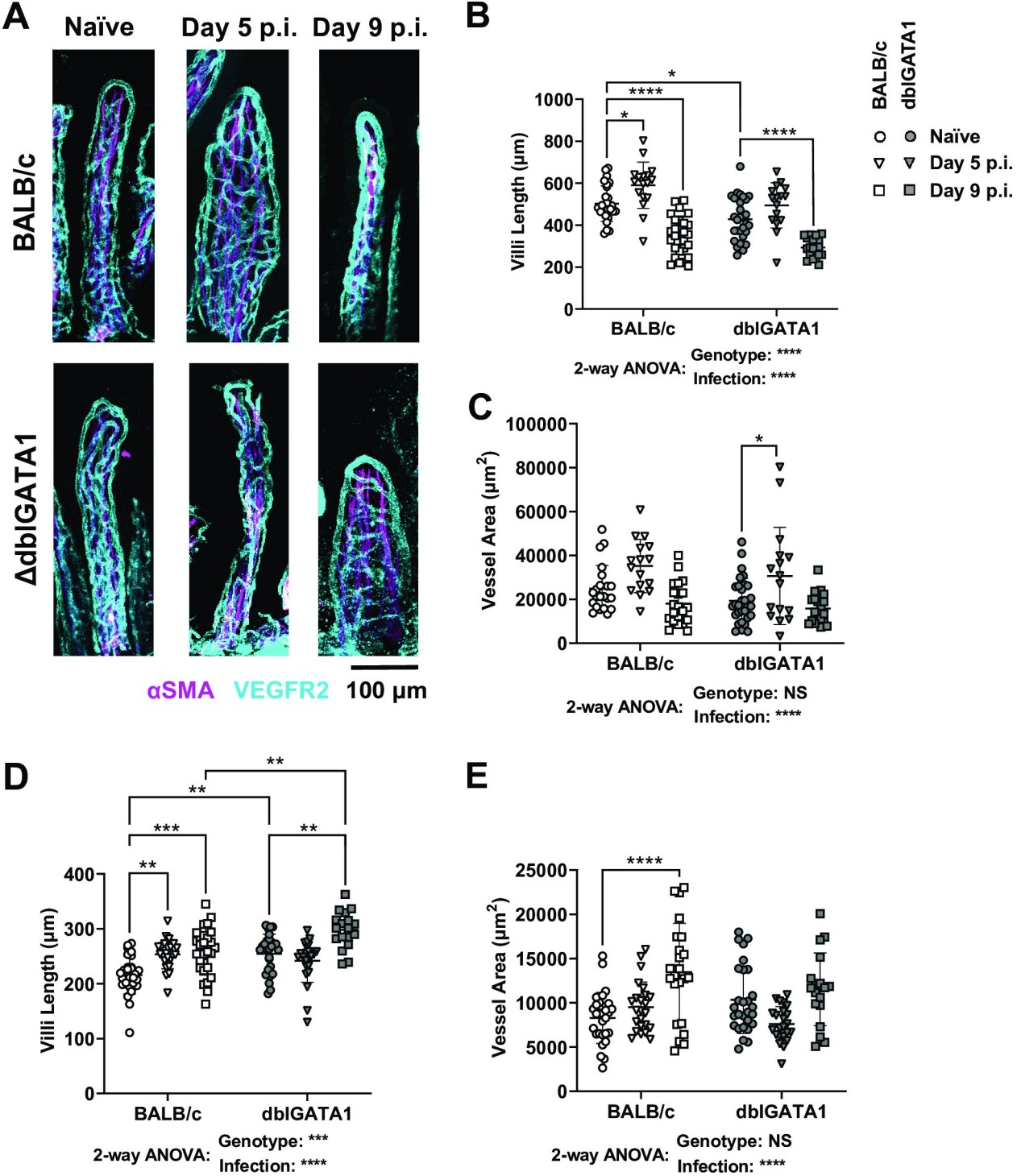

**Fig 2. Impact of Nb infection on jejunal villous architecture in BALB/c and dblGATA1 mice.** dblGATA1 mice and wildtype BALB/c controls were infected with 400–500 L3 Nb and sacrificed at the indicated days p.i. Control groups of naïve mice were included. (A) Jejunal whole mounted tissue from naïve and Nb infected mice stained with αSMA (magenta) and αVEGFR2 (cyan) antibodies. Images were taken as z-stakes that capture entire villious structures and are presented as a composite image. Jejunal villi length (B) and villi surface area (C) and ileal villi length (D) and villi surface area (E) were measured using FIJI (Image J). Villi length was determined using the line tool and surface area was calculated as the area enclosed by VEGFR2 staining using the polygon tool. Data in A show representative images of single villi from each group. Data in B—E are one representative of two experiments with symbols represent individual villi, using n = 3–5 animals per group with 2–8 villi per animal. All data are shown as mean ± SEM and significance determined using a two-way ANOVA with Tukey's post-hoc analysis.

decrease in both the length and surface area of jejunal villi at day 9 p.i. consistent with villous atrophy (Fig 2A–2C). Interestingly, we noticed an increase in the villous length (Fig 2D) and surface area (Fig 2E) in the ileum at both day 5 and 9 p.i. Such changes in the distal small intestine may be a compensatory mechanism by the host to increase nutrient absorption in the face of jejunal villous atrophy. Consistent with our previous report eosinophils promoted the maintenance of villous structure in naïve mice [26] (Fig 2A & 2B & 2D), but their presence or absence had no impact on Nb-induced alterations to villous architecture in the jejunum (Fig 2A–2C).

## N. brasiliensis infection results in minor changes to the neuronal density but does not alter the neurochemistry of enteric neurons

To examine the role of the ENS in Nb-induced small intestinal hypermotility we performed imaging studies to assess the density of neuronal cell bodies using antibodies directed against the pan neuronal marker HuC/D. Neuronal structure or density was determined by determining the number of HuC/D$^+$ cells within a defined tissue area for the myenteric plexus, or by calculating the number of HuC/D$^+$ cells per submucosal ganglion (a cluster of neuronal cell bodies) in the submucosal plexus (illustrated in Fig 3A). Nb infection slightly altered neuronal structure in the myenteric plexus as indicated by a reduction in neuronal density (Fig 3B & 3C). Although these data indicate Nb infection impacts the ENS in the myenteric plexus, it is possible that the increased intestinal length noted in infected animals contributes to this (Fig 1B), rather than it being completely explained by a loss of neurons. In keeping with this hypothesis, we did not observe an impact of infection on the average number of neuronal cells per ganglion in the submucosal plexus (Fig 3D & 3E). Based on our hypothesis that eosinophil-neuron interactions may contribute to ENS function, we included both BALB/c and dblGATA1 mice in our experiments. However, the presence or absence of eosinophils did not affect any of the parameters measured (Fig 3).

We next investigated whether infection influenced the expression of the major neurochemical markers of the ENS. In the myenteric plexus, the proportions of neurons expressing either or both calbindin (CalB) and calretinin (CalR), two calcium-binding proteins that are expressed by the major subpopulations of excitatory neurons, were unchanged following Nb infection and not impacted by eosinophil deficiency (Fig 4A & 4B). We also determined the impact of Nb infection on neuronal nitric oxide synthase (nNOS), a well-defined marker of inhibitory motor neurons and some interneurons. As expected, most nNOS$^+$ neurons did not co-express CalB and Nb infection did not alter the proportion of either nNOS$^+$ or CalB$^+$ single positive neurons (Fig 4C & 4D). We also observed a small subpopulation of CalB$^+$/nNOS$^+$ neurons of unknown function, however these were not impacted by infection. Eosinophil deficiency resulted in a slight reduction in the proportion of CalB$^+$ neurons in this experiment (Fig 4C & 4D). But this is unlikely to be meaningful as the same subpopulation, identified in our earlier assessment of CalB and CalR expression, was not altered by eosinophil deficiency (Fig 4B versus 4D). Additionally, we examined the axonal connectivity between myenteric ganglia using calbindin staining and did not observe any gross changes (S1 Fig). Finally, in the submucosal plexus, neither Nb infection nor eosinophil deficiency influenced the expression of vasoactive intestinal peptide (VIP) and choline acetyltransferase (ChAT), which are markers of the two major classes of submucosal neurons respectively and appear to account for the entire population [30] (Fig 4E & 4F). A smaller subpopulation of VIP$^+$/ChAT$^+$ neurons was also unaffected (Fig 4F). Overall, these data indicate that neither Nb infection, nor eosinophils, have an impact on small intestinal ENS structure or the expression of tested neurochemical markers.

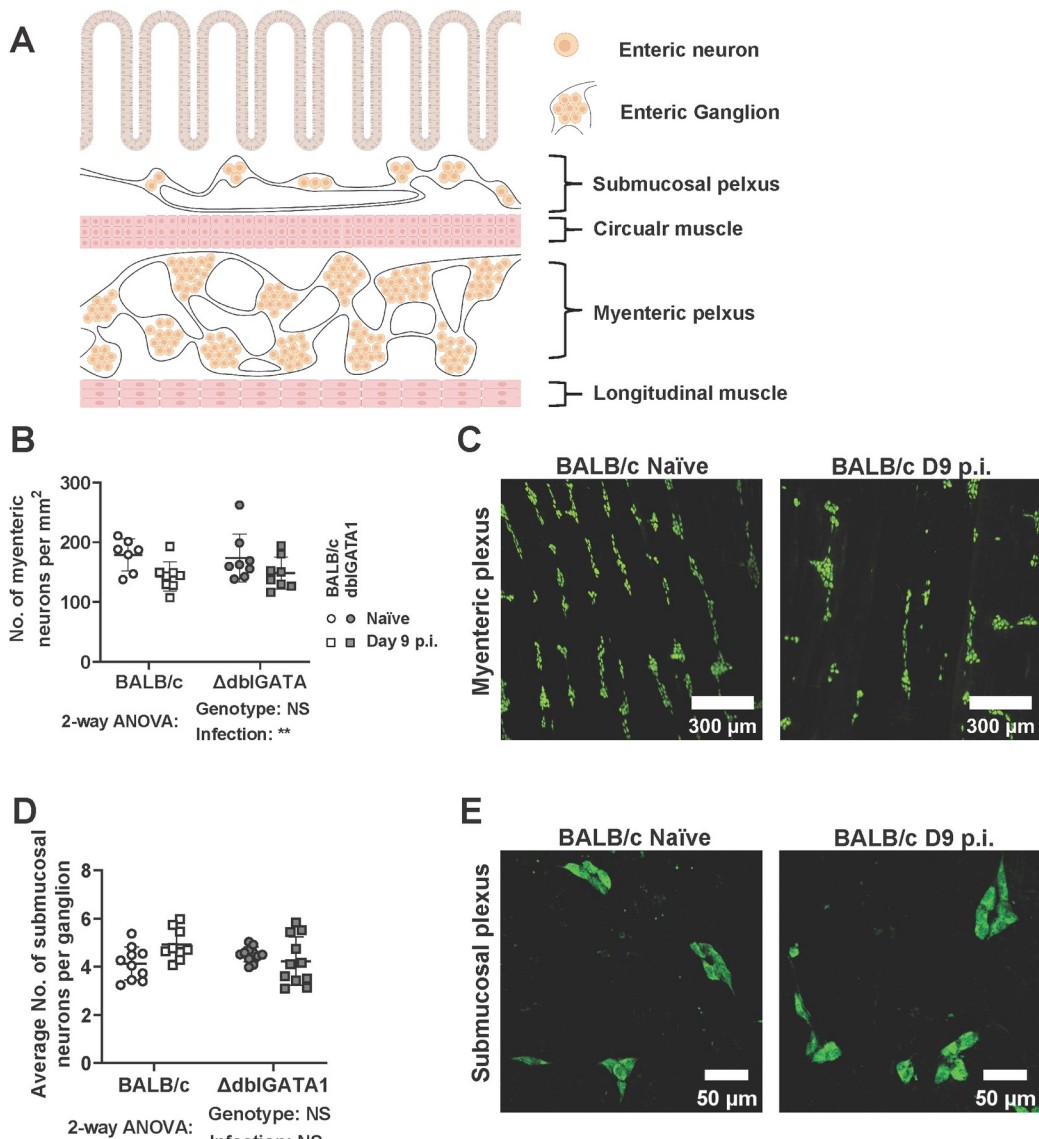

**Fig 3. Impact of Nb infection on neuronal structure in the submucosal and myenteric plexus.** dblGATA1 mice and wildtype BALB/c controls were infected with 400–500 L3 Nb and sacrificed day 9 p.i. Control groups of naïve mice were included. Myenteric and submucosal plexus layers were collected as detailed in Materials and Methods and stained with a pan neuronal marker HuC/D. (A) Schematic illustration of the neuroanatomy of the ENS inside the small intestine created with BioRender.com. The size of cells and tissue structures is not to scale. (B) Myenteric neuronal cell density was calculated by counting cell bodies and presented as number/mm². (C) Representative images of myenteric plexus stained with HuC/D from a single naïve and infected BALB/c mouse. (D) Number of neurons per ganglion in the submucosal plexus. (E) Representative images of submucosal plexus stained with HuC/D from a single naïve and infected BALB/c mouse. Symbols in A and C represent individual animals with data pooled from 2–3 separate experiments, n = 9–11 per group for submucosal plexus and n = 7–8 per group for myenteric plexus. All data are shown as mean ± SEM and significance determined using a two-way ANOVA with Tukey's post-hoc analysis.

## Eosinophils are present in the submucosal, but not myenteric, plexus of the small intestine

We had previously determined that eosinophils interact with axons in the jejunal villi [26], and Ahrends and Aydin et al. [31] reported that eosinophils influence myenteric neuronal survival during helminth and viral co-infection. Although our current data indicate that eosinophils do not

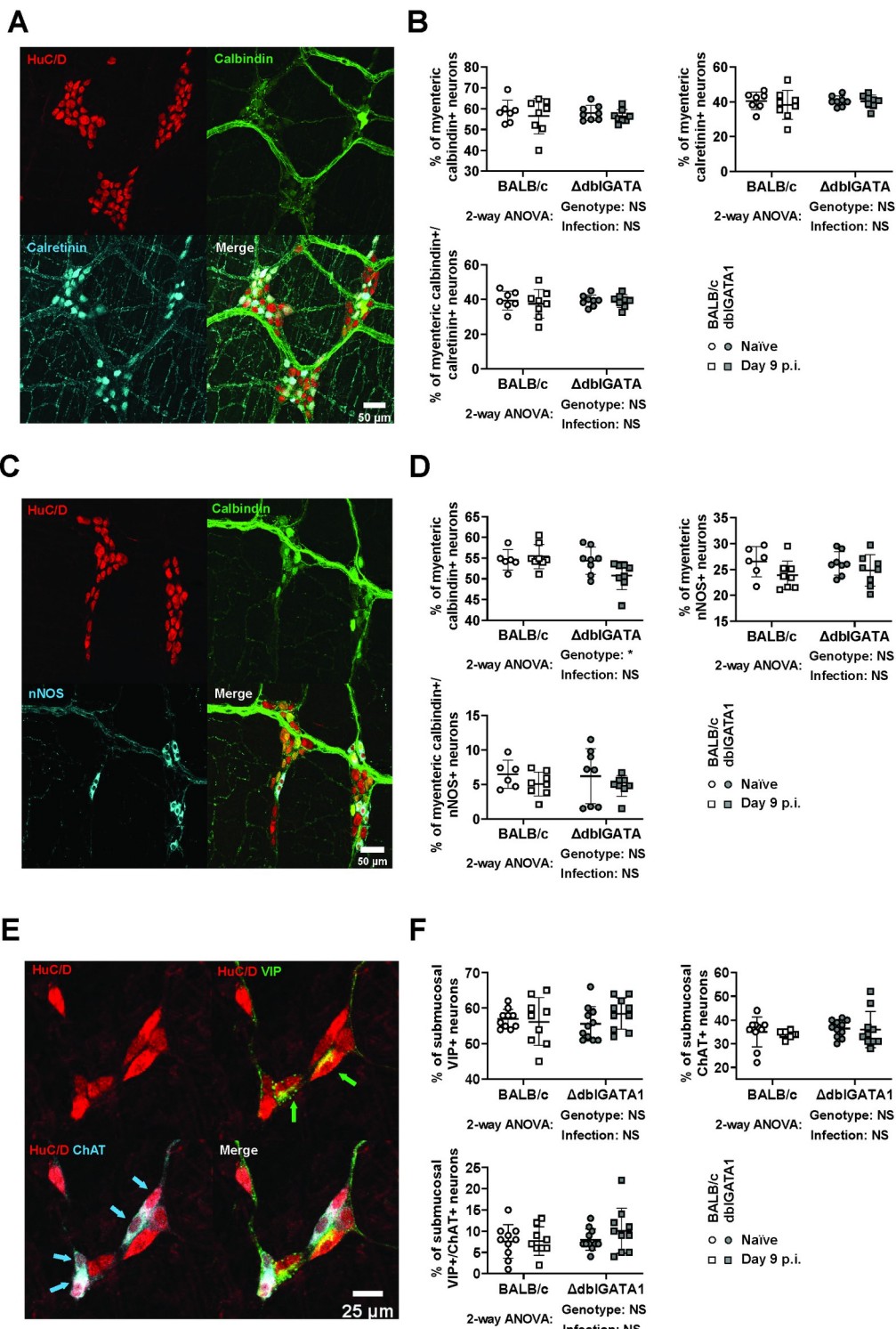

**Fig 4. Impact of Nb infection on neurochemical marker expression in the submucosal and myenteric plexus.** dblGATA1 mice and wildtype BALB/c controls were infected with 400–500 L3 Nb and sacrificed day 9 p.i. Control groups of naïve mice were included. Myenteric and submucosal plexus were collected as mentioned in Materials and Methods and stained for the indicated neurochemical markers using antibodies. (A, C&E) Representative images are shown from a single naïve BALB /c animal for myenteric plexus stained with antibodies directed against (A) HuC/D (red), CalB (green), and CalR (cyan) or (C) HuC/D (red), CalB (green), and nNOS (cyan) and for (E) submucosal plexus stained with antibodies directed against HuC/D (red), VIP (green), and ChAT (cyan). (B, D &F) The percentage

of total neurons expressing the indicated neurochemical markers was calculated by counting at least 180 HuC/D$^+$ cell bodies in myenteric plexus and least 100 HuC/D$^+$ cell bodies in submucosal plexus from each animal. Symbols represent individual animals with data pooled from 2–3 separate experiments, n = 9–11 animals per group for submucosal plexus and n = 7–8 animals per group for myenteric plexus. All data are shown as mean ± SEM and significance determined using a two-way ANOVA with Tukey's post-hoc analysis.

impact small intestinal motility or ENS structure during homeostasis or Nb infection, we wondered whether these cells form contacts with neurons in the myenteric or submucosal plexus. No eosinophils were present in the myenteric plexus (S2A Fig), even after Nb infection (S2B Fig). However, eosinophils could be found in the submucosal plexus, both in naïve mice and after Nb infection, with some cells located in close proximity to neuronal cell bodies, with a distance < 1 μm (S2C & S2D Fig). To further validate these findings, we performed intestinal wholemount imaging and showed that no eosinophils could be found in the muscle wall (S2E Fig). Lastly, we confirmed the specificity of the anti-SiglecF antibody by staining the submucosal plexus from dblGATA1 mice (S2F Fig). These data indicate that if eosinophils engage in neuroimmune crosstalk they must do so through interactions formed with axons present in the lamina propria and/or neuronal cell bodies present in the submucosa, but not in the muscularis of the small intestine.

### *N. brasiliensis* infection only results in minor alterations to ENS function

Although Nb infection had little effect on ENS structure or neurochemistry, it is possible that infection did alter neurogenic-driven muscle contractions not detected by our analysis. To more fully interrogate possible alterations, we performed video imaging of contractions occurring in excised jejunum equilibrated to physiological conditions in a horizontal organ bath. Recordings of intestinal contractions were made and analysed using a previously established method [32,33] (S3 Fig) and measurements made with and without addition of DMPP, a nicotinic acetylcholine agonist that can directly activate enteric neurons but not muscle cells. Similarly to previous reports [32], most neurogenic-driven contractions in our experiments were initiated from the distal jejunum and propagated proximally (Fig 5A). Neither the frequency of neurogenic initiations or peristaltic contractile complexes (PCCs) were changed by Nb infection, and this was true when examining tissues with or without addition of DMPP (S4A, S4B Fig). Infection had no impact on the amplitude (Fig 5B), duration (S4C Fig), or velocity (Fig 5C) of neurogenic-driven muscle contractions without DMPP (S4C Fig, 4B & S4C Fig). However, DMPP stimulation did increase contraction amplitude in tissue from infected but not from naïve mice (Fig 5C), and comparison of the contraction parameters with or without DMPP stimulation revealed an effect of DMPP on the amplitude (Fig 5B) and velocity (Fig 5C) but not duration (S4C Fig) of contractions in tissues from infected mice. Taken together these data suggest that the pattern of neurogenic-driven muscle contractions, which are generated by the ENS, are altered by Nb infection. However, these alterations are modest and only become evident in measurements of the DMPP-activated response, suggests a change in synaptic transmission consistent with previous studies of the effect of nematode infections on enteric neurons [34]. To validate that the observed effect of DMPP was indeed restricted to the ENS, we further confirmed that the slow wave frequency, which results from myogenic-driven contractions of circular muscle, was not altered by either infection or addition of DMPP (S4D Fig).

### Increased intestinal motility in response to secondary *N. brasiliensis* infection is largely driven by smooth muscle hypertrophy

In the absence of a major change in neurogenic-driven muscle contractions, we hypothesized that infection-induced hypermotility may be driven by myogenic alterations, such as altered

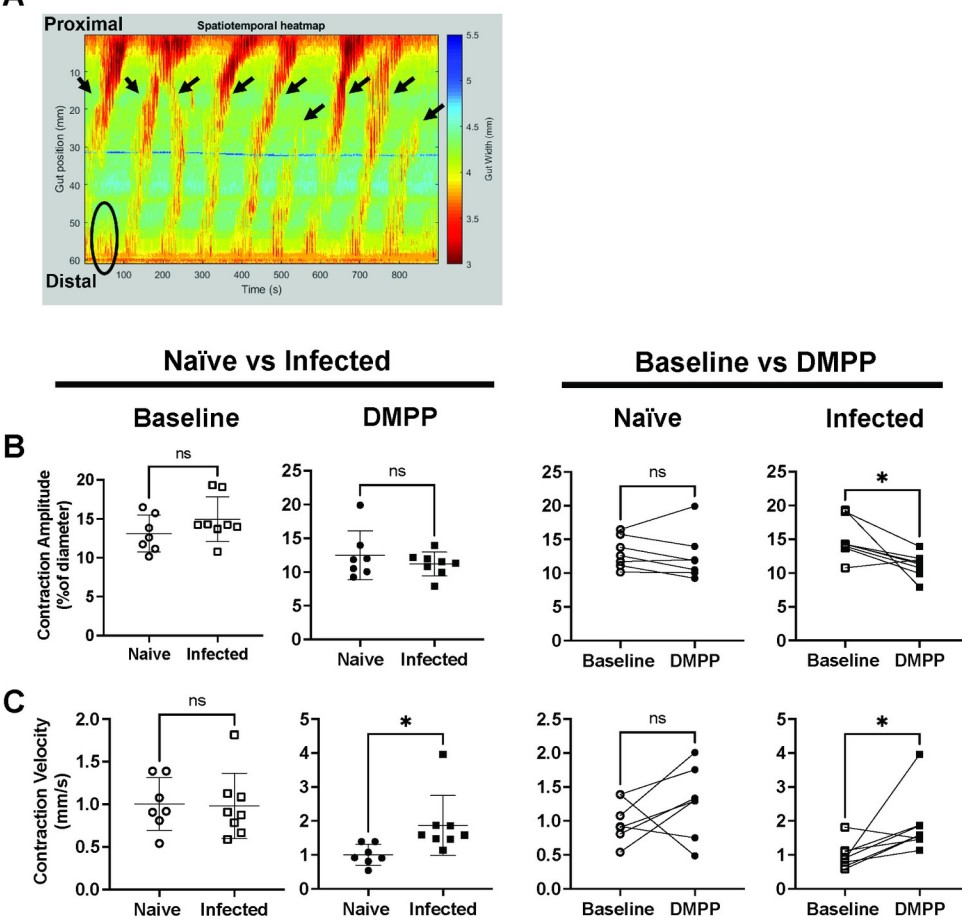

**Fig 5. Nb infection does not influence neurogenic-driven muscle contraction in the jejunum.** BALB/c mice were infected with 400–500 L3 Nb and sacrificed day 6–9 p.i. Control groups of naïve mice were included. Jejunal contractile activities were recorded under physiological conditions in a horizontal organ bath using a video camera, and data analysed with in-house software Scribble 2.0 and Analyse 2. A) An example image of spatiotemporal heatmap showing jejunum contractions at baseline. X-axis on the left indicates the position of the tissue, y-axis indicates time. Each color-coded pixel indicates jejunum diameter (based on the scale on the left) at each position at a specific time, thus each warm coloured band indicates a contraction. All contractions that started from the distal half of the jejunum are recorded as initiations (circles and arrow heads), and initiations that propagated through more than 50% of the tissue are recorded as PCCs (arrow heads). Analysis of the spatiotemporal heatmap provided information on PCC contraction duration and velocity, contraction amplitude, and slow wave frequencies (see also S2 Fig). (B&C) Comparison of (B) contraction amplitudes and (C) contraction velocities recorded in tissues from naïve or infected mice under baseline conditions or after DMPP application. Open and closed symbols indicate recordings taken during baseline conditions (open) or after DMPP application (closed). Symbols represent individual animals (n = 7–8 per group), with each experiment including tissues from one naïve animal and one infected animal kept within the individual chambers of the same organ bath. Tissues that ceased to contract before the end of the recording were excluded from analyses. Data are shown mean ± SEM and significance determined using unpaired student T-tests comparing naïve and infected mice, and paired student T-tests for comparing baseline and DMPP.

muscle mass or reactivity to neurotransmitters. To test these hypotheses, we determined muscle thickness in tissue sections from the upper jejunum and measured the strength of longitudinal muscle contractions in an organ bath as previously described [32]. Infection resulted in a significant increase in the thickness of the muscularis tissue, with hypertrophy evident in both the circular and longitudinal muscle layers (Fig 6A–6C). This increased muscle thickness could be directly correlated with observations of contraction strength made in a multi-chamber organ bath using jejunal tissues mounted longitudinally under tension and connected to

force transducers. In this assay tissues from infected mice had normal contraction frequencies but increased contraction strength as compared to naïve mice (Fig 6D & 6E). The type 2 cytokines, IL-4 and IL-13, are known to drive muscle hypertrophy and hyperresponsiveness [4,35–38] and as expected Nb infection led to significantly higher jejunal IL-4 and IL-13, but not IFN-γ, gene expression (S5A–S5C Fig). Type 2 cytokines are also known to alter muscle responsiveness to acetylcholine [5,39] and we were able to confirm a dose-dependent increase in contraction strength following acetylcholine stimulation of tissues from Nb infected mice (Fig 6F & 6G). In keeping with our earlier data, the presence or absence of eosinophils had little impact on this response (Fig 6). Of note, tissue taken from the distal ileum of Nb infected mice exhibited parallels with that from the jejunum with unaltered contraction frequencies (S6A Fig), but increased contraction strength (S6B Fig) and responsiveness to acetylcholine (S6C & S6D Fig). This suggests the effects of infection on muscle hypertrophy are not restricted to the location of the worms, which typically reside in the duodenum. These data support a central role for myogenic, rather than neurogenic, changes in helminth induced small intestinal hypermotility.

## Increased intestinal motility in response to secondary *Heligmosomoides polygyrus* infection is also associated with smooth muscle hypertrophy

Unlike Nb, most intestinal helminths cause chronic infections in natural settings. We thus additionally investigated neurogenic and myogenic alterations occurring in response to primary and secondary Hp infection [7]. While primary Hp infections did not alter small intestinal length or transit, mice with secondary Hp infections had significantly longer small intestines and faster transit (Fig 7A & 7B). This increase of small intestinal transit occurred alongside reduced worm burdens observed in secondary compared to primary infected mice (Fig 7C), which further supports the hypothesis that intestinal hypermotility promotes worm expulsion [2,3]. We next determined smooth muscle thickness and found longitudinal and circular muscle hypertrophy in both primary and secondary Hp-infected mice (Fig 7D & 7E). To investigate the potential involvement of the ENS in Hp-induced hypermotility, we performed neuroanatomical and neurochemical analysis of the myenteric plexus akin to that performed following Nb infection. Both primary and secondary Hp infection led to significant reductions in myenteric neuronal density (S7A & S7B Fig). As for Nb the reduced neuronal density is likely to partially reflect growth of the small intestine noted in the increased intestinal length (Fig 7A), but it may also reflect neuronal death. In terms of neurochemistry, the proportions of myenteric neurons expressing either CalB or CalR, which are putative excitatory neurons, remained unchanged following infection (S7C & S7D Fig). However, there was a small but significant reduction of the proportions CalB+/CalR+ neurons in secondary Hp-infected mice (S7C & S7D Fig). Given smooth muscle hypertrophy occurred in both primary and secondary infection, the observed alterations in CalB+/CalR+ neurons may play a role in the faster transit time observed following secondary infection. In contrast, the proportion of inhibitory neurons expressing nNOS were not impacted by primary or secondary infection (S7E & S7F Fig). We noticed a significant reduction of CalB+ neurons in primary Hp-infected mice (S7E & S7F Fig) but this is likely to be a type 1 error as we did not find any alterations to this subpopulation in our previous analyses of CalB and CalR expression (S7C & S7D Fig). Neither primary nor secondary Hp infection impacted the proportion of neurons expressing a CalB+/nNOS+ phenotype (S7E & S7F Fig). Together, our results indicate that Hp infection results in marked smooth muscle hypertrophy and reduced neuronal density in the myenteric plexus. Subtle, but potentially important alterations to neuronal neurochemistry were noted following secondary Hp infection.

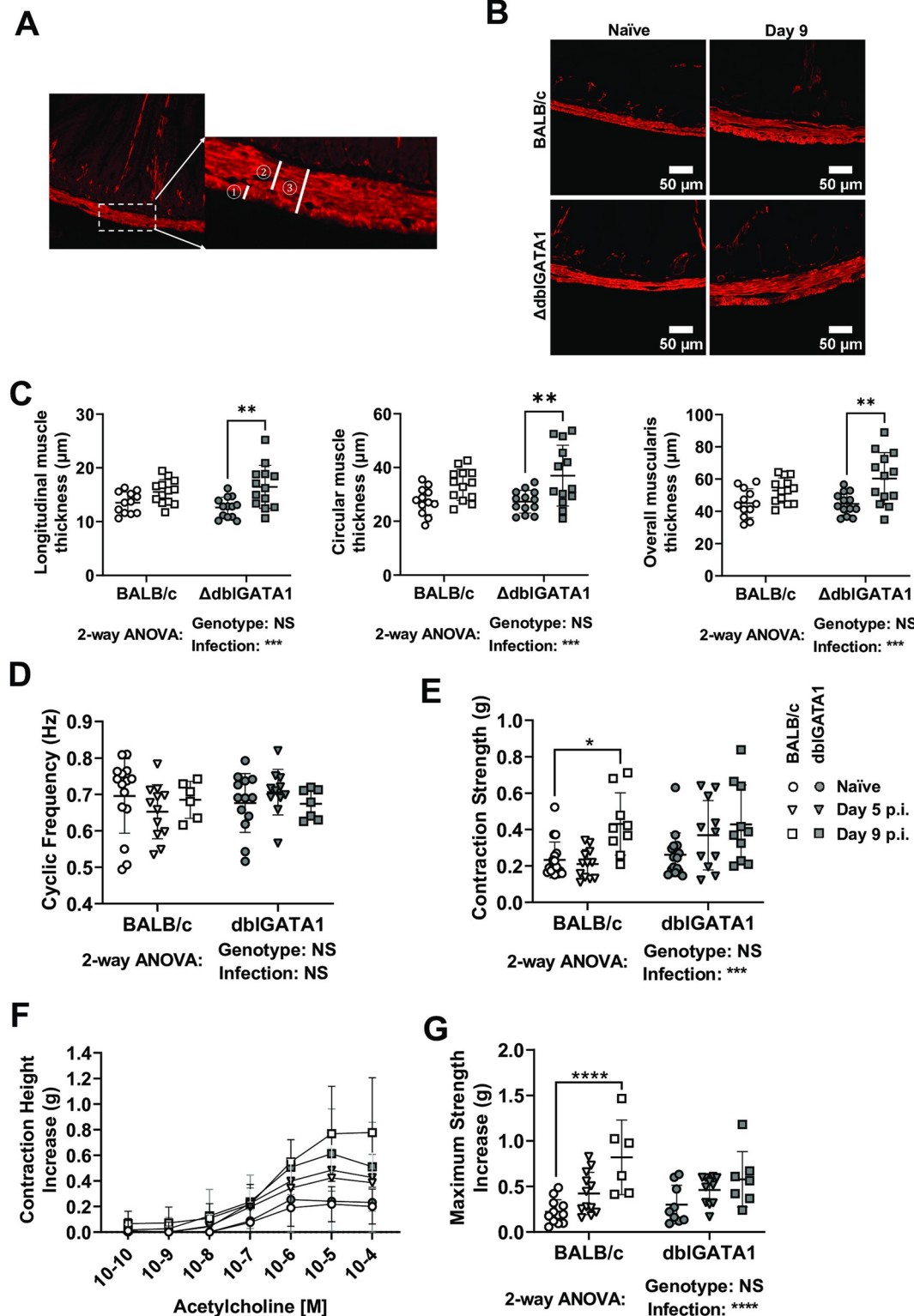

**Fig 6. Nb infection drives intestinal smooth muscle hypertrophy and hypercontractility.** dblGATA1 mice and wildtype BALB/c controls were infected with 400–500 L3 Nb and sacrificed at the indicated days p.i. Control groups of naïve mice were included. (A&B) Representative images are shown for jejunal cross-sections from single naïve or Nb infected BALB/c and dblGATA1 mice stained with antibodies directed against the muscle cell marker SMA. Muscle thickness was measured using the line tool from FIJI (Image J) for the ① longitudinal ② circular and ③ total muscularis layers. (C) Muscle thickness

was determined as shown in A). 40 measurements were made per muscle layer from each animal, with symbols representing the mean of individual animals. (D-G) jejunal tissues were mounted longitudinally within an organ bath as described in the Materials and Methods and (D) contraction frequency and (E) contraction strength recorded under baseline conditions. Thereafter the same tissues were stimulated using incremental dosages of acetylcholine and (F) the dose-dependent change in contraction strength or (G) the maximum increase in contraction strength. Symbols represent individual animals pooled from 2 independent experiments (n = 7–13 animals/group). Data are shown mean ± SEM and significance determined using a two-way ANOVA with Tukey's post-hoc analysis.

## Discussion

Enteric pathogens are well known to impact the ENS, with *Salmonella Typhimurium* causing loss of neuronal bodies in the murine myenteric plexus [10,31]; neurotrophic viruses, such as West Nile virus, able to target enteric neurons and impar intestinal motility [14–16]; and toxins from *Cholera difficile* inducing prolonged hyperexcitability of submucosal neurons [11,13], thereby contributing to fluid hypersecretion during cholera infection. Helminths also impact the ENS with *Schistosoma japonicum* infection of pigs altering the proportion of enteric neurons expressing Substance P or VIP [17], and *Trichenalla spiralis* increasing neuronal excitability and synaptic plasticity in the ENS of guinea pigs [34,40]. Intestinal hypermotility in hookworm infected mice has been predicted to involve the ENS [5]. This prompted us to further investigate the ability of Nb infection to regulate ENS function.

To our surprise Nb infection resulted in no, or only minor, alterations to enteric neuron densities and neurochemical markers tested in the submucosal and myenteric plexus of the small intestine. Infection also had little impact on the frequency, duration or velocity of neurogenic-driven muscle contraction patterns in the small intestine. However, application of the nicotinic agonist, DMPP, did reveal small differences between uninfected and infected tissues indicating that neurons in these tissues can respond differently to cholinergic stimulation via synaptic receptors. It was previously reported that longitudinal muscle strips from Nb-infected mice have strong contractile responses upon *ex vivo* electric field stimulations (EFS) that only activate enteric neurons. While these data strongly suggest involvement of neurons, the observed increases in contraction strength could also result from changes in muscle sensitivity to neurotransmitters rather than in changes to the nerves themselves [5]. The same publication reported that muscles from Nb infected mice exhibit increased sensitivity to acetylcholine or substance P driven contraction, and that contractions were tetrodotoxin sensitive and hence dependent on neural activity [5]. A possible impact of Nb infection on neuronal excitability in response to acetylcholine acting on nicotinic receptors is in line with our data showing that DMPP stimulation of tissue from naïve or Nb infected mice yielded different outcomes for neurogenic-driven muscle contractions. Despite these observed alterations in neuronal responses to nicotinic stimulation, the fact that Nb infection did not impact ENS neurochemistry or baseline neurogenic-driven muscle contractions suggests that altered ENS activity is unlikely to be the main determinant of the increased small intestinal hypermotility that occurs in response to infection. Importantly, our experiments were only performed using an *ex vivo* motility model. It is therefore possible that individual neuronal excitability is altered during Nb infection, but that the changes are not pronounced enough to alter neurogenic-driven contractile patterns at baseline. Our study is also limited by only considering the impact of intrinsic components of the ENS on hypermotility, as it remains possible that extrinsic neural innervation is altered by Nb infection and contributes to hypermotility. However, we lacked an appropriate *in vivo* model that would allow us to study the impact of such innervation on motility.

In our experiments we observed that Nb infection increases muscle contraction strength under baseline conditions and repeated previous observations [41] that longitudinal intestinal

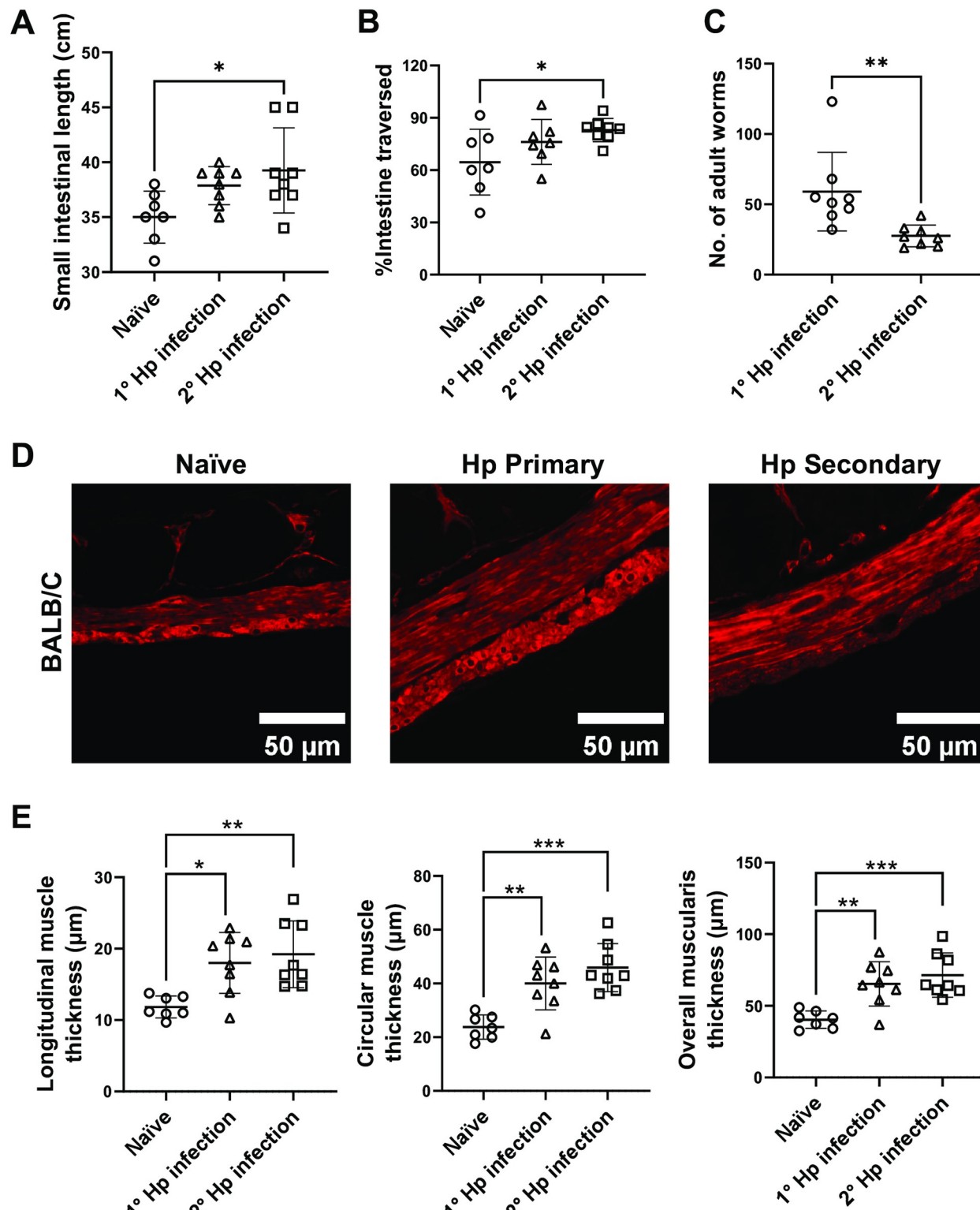

**Fig 7. Hp infection alters small intestinal motility and smooth muscle hypertrophy.** BALB/c mice were infected with 200 L3 Hp at day 0 for the primary infection group or day 0 and day 37 for the secondary infection group and were sacrificed day 14 p.i. or day 51 p.i., respectively. Control groups of naïve mice were included. (A) Length of the entire small intestine was determined. (B) Intestinal motility was determined using a carmine red dye transit assay. Dye was administered by oral gavage 30 mins prior to sacrifice and motility determined as the distance travelled by dye (as a percentage of total small intestine length) at the time of sacrifice. (C) Number of luminal adult Hp worms were counted. (D)

Representative images are shown for jejunal cross-sections from a single naïve, primary Hp infected, or secondary Hp infected BALB/c mice stained with antibodies directed against the muscle cell marker SMA. (E) Longitudinal muscle, circular muscle, and overall muscularis thickness was determined as detailed in Method. Symbols represent individual animals pooled from 2 independent experiments (n = 7–8 animals/group). Data are shown mean ± SEM and significance determined using a one-way ANOVA with Tukey's post-hoc analysis.

segments from Nb infected mice exhibit increased sensitivity and contraction responses following the addition of acetylcholine to an organ bath. Alongside our observations of small intestinal muscle hypertrophy these findings indicate that myogenic-driven changes likely explain the small intestinal hypermotility observed in response to Nb infection. As mentioned earlier the type 2 cytokines IL-4 and IL-13 cause smooth muscle hyperplasia/hypertrophy and upregulate muscarinic acetylcholine receptor M3 [4,35,39], which is expressed by intestinal smooth muscle [41]. In keeping with a key role for M3 receptors, mice lacking these receptors do not exhibit increased muscle contractility following Nb infection and display a reduced type 2 immune responses and impaired worm expulsion [39,42].

We further investigated the potential involvement of the ENS using Hp, a chronic murine helminth infection [7]. Like Nb, Hp drove significant increases in muscle thickness, consistent with a previous report showing that longitudinal muscle strips from secondary Hp-infected mice exhibit stronger contractions in response to EFS and acetylcholine [5]. Both primary and secondary Hp infection led to significant reduction of neuronal density. However, only minor neurochemical changes were observed, and these were restricted to secondary Hp infection.

Whilst we know eosinophils are dispensable for the expulsion of adult Nb following primary infection [43,44] (Fig 1), they do promote host resistance to repeated Nb infection [19,44]. Based on recent findings that enteric eosinophils promote intestinal homeostasis in response to microbial colonisation [26], and protect the colon from excessive inflammation in response to chemical damage [45] we hypothesised that they also contribute to adaptation of the intestine in response to Nb infection. Although we confirmed an important role for eosinophils in maintaining small intestinal villous morphology under homeostatic conditions, we found no evidence that they provide protection against hookworm-induced villous atrophy. We did note however that both wildtype and eosinophil deficient mice exhibited increased villi length in the ileum following infection, perhaps to compensate for the reduced villous surface in the upper small intestine to maximise nutrient uptake.

Enteric neuroimmune interactions are important in regulating intestinal function in both homeostasis and disease [46] and eosinophils can promote axonal branching in the skin and airways [47, 48]. Yet it remains unknown whether eosinophils impact on, or communicate with, enteric neurons. We investigated whether eosinophils could contact enteric neurons by immunohistochemical staining and determined homeostatic small intestinal motility and ENS structure in mice with or without intestinal eosinophils. Our data showed that, whilst eosinophils can make close contact with submucosal neurons, they are not found in the myenteric plexus of either healthy or hookworm infected mice. The absence of eosinophils in the jejunal myenteric plexus is in line with our finding that eosinophil deficiency does not alter small intestinal transit. The presence of eosinophils also had no influence on the structure or neurochemistry of the submucosal or myenteric plexus, suggesting that eosinophils are not required for the development and maintenance of the small intestine ENS. We have previously reported that eosinophil deficiency influences whole gut transit, which measures a combination of small and large intestinal motility [26], with transit through the large intestine typically taking much longer than transit through the small intestine [49,50]. Based on our current data it is likely that eosinophils influence motility exclusively within the colon and further experiments will be needed to determine exactly how this occurs. Regarding helminth infections, it remains

possible that eosinophils impact pain perception during infection as it has been reported that eosinophils promote axonal branching, especially for those neurons with sensory function [47,48]. Lastly, our finding of eosinophils contacting jejunal submucosal neurons suggests that eosinophils regulate fluid secretion and vasodilation, although this hypothesis was not tested in the current manuscript due to technical limitations.

In summary, we show that Nb infection leads to a dramatic small intestinal hypermotility, and that this response directly correlates with worm expulsion. We provide evidence that hypermotility during Nb infection is driven largely by changes to muscle strength and responsiveness rather than changes to the ENS. Alterations of smooth muscle are also apparent after Hp infection, although it is likely the ENS also contributed to efficient expulsion of this helminth. Intimate contacts were observed between eosinophils and submucosal neurons, but eosinophils were not present in the myenteric plexus and their absence did not impact enteric neuronal structure and neurochemistry in healthy or infected intestinal tissue. We suggest that alterations to muscle may also occur in response to other infectious and inflammatory conditions and that myogenic alterations should be investigated alongside neurogenic alterations in these conditions.

## Materials and methods

### Ethics statement

All procedures were approved by Alfred Research Alliance Animal Ethics Committee, Melbourne, Australia (Ethics number: E/1828/2018/M and E/8234/2022/M).

### Animals and tissue collection

BALB/c and eosinophil deficient ΔdblGATA1 mice were bred at Monash Animal Research Platform and housed in the Monash ICU facility or Level 7 Baker tower after arrival. Mice with different genotypes were age (8–14 weeks) and sex-matched, and their beddings were mixed twice a week for at least three weeks to balance microbiota. A similar number of naïve mice were included for each experiment and each time point as controls. Both male and female mice were used, except for video imaging experiment for jejunal motility where only male mice were used to avoid differences in the oestrus cycle. All mice were humanely killed by cervical dislocation except for performing villous wholemount staining where they will killed by anaesthesia then exsanguination. The first 5 cm of small intestines (duodenum) were discarded. The tissues were cut in half, and we defined the most proximal 5 cm of tissue from each section as jejunum and ileum, respectively.

### Helminth and infection

For Nb infections, stage 3 Nb larvae (L3) were either recovered from Lewis rats [51] or shipped from New Zealand. L3 Nb larvae were kept on Petri dishes with wetted filter paper at 25˚C. Upon infection, mice were injected with 400–500 L3 Nb in 200 μL autoclaved distilled water subcutaneously.

For Hp infections, stage 3 larvae (L3) were generated using published protocols [51] and were kept in distilled water at 4˚C. For animals that received one infection, mice were orally gavaged with 200 L3 Hp and humanely culled on day 14 p.i. For animals that received two infections, mice were orally gavaged with 200 L3 Hp for the primary infection. On day 28 and day 30 p.i, mice were orally gavaged with an anti-helminthic cobantril (10mg/kg), to facilitate the clearance of initial infection. Mice then received the secondary infection on day 37 (200 L3 Hp, oral gavage), and were humanely culled on day 51 (day 37+14).

## Carmine red transit assay

Carmine red dye (3 mg/mouse, Sigma Aldrich C1022) was used in conjunction with the thickening agent methylcellulose (0.25 mg/mouse, Sigma Aldrich 274429) and mixed in PBS to mimic a food bolus. Mice were gavaged orally with 50 or 100 μL carmine red/methylcellulose solution, and small intestines were collected 30 minutes after. The distance the dye had travelled, and the total intestinal length was recorded. Intestinal transit was calculated by dividing the distance travelled by the total length of the small intestine.

## Worm counts

Intestinal worm numbers were counted after performing the Baermann assay. Intestines were opened lengthways and cut into a few pieces carefully. Pieces of intestine were then wrapped in a single layer of cheesecloth and submerged in PBS in a 50ml falcon tube. Falcon tubes were placed in a water bath at 37˚C for at least 1 hour to allow worms migration along the temperature gradient. Solutions containing worms were collected from the bottom of the tubes with a transfer pipette and counted on a gridded plate under a dissection microscope.

## Villous wholemount staining

The protocol for villous wholemount staining was previously described [29]. In brief, mice were anaesthetised with Ketamine (200mg/kg, Mavlab) and Xylazine (20mg/kg, Ilium) until they were unresponsive to the foot pinch test. Mice were perfused through the left ventricle with 10ml of PBS, followed by 10ml of 4% paraformaldehyde. Jejunum and ileum were collected, cut open, and pinned loosely in silicon Petri dishes in PBS. Tissues were fixed with a fixation buffer (0.5% paraformaldehyde (PFA), 15% Picric, and 100 mM of phosphate buffer in distilled water (pH 7) overnight. Tissues were washed in PBS for 5 x 10 minutes and then incubated in 10% sucrose solution in PBS for 3 hours. The washing steps were repeated, and the tissues were incubated with sucrose/glycerol (20%/10%) solution in PBS overnight. Tissues were washed once the next day before being immersed in PBS containing 0.1% sodium for further processing. All washing and incubating steps were done at 4˚C on a circular shaker.

To stain the intestinal segments, around 1 cm samples were cut and pinned to a 12-well silicon-based plate. Samples were submerged in blocking buffer (5% Donkey serum, 0.5% bovine serum albumin, 0.3% Triton X-100, 0.1% NaN3 in PBS) for a minimum of 2 hours. Tissues were then incubated in primary antibodies (in the blocking buffer). Tissues were washed 5 x 1 hour with washing buffer (0.03% Triton X-100 in PBS) before being incubated in primary antibodies (in blocking buffer) overnight. Samples were then washed 10 times for 30min in washing buffer. Once washed, samples were then immersed in 4% PFA for 36–48 hours to further fixate. All washing and incubating steps were done at 4˚C on a circular shaker.

After being washed with PBS, tissues were cut into 1–3 villi thick strips under a dissection microscope. Strips were placed on superfrost microscope slides containing 2 spacers between the slide and cover slip to avoid crushing villi. Samples were mounted using fluorescent mounting medium (DAKO), and the coverslip was added. Samples were then stored in dark at 4˚C until required for microscopy.

## Plexus wholemount staining

Jejunums were carefully collected without damaging the muscularis. The excess mesentery was removed, and the tissue was opened along the mesentery boarder using fine spring scissors. Tissue was then stretched and pinned to a PBS-filled Sylgard Petri dish under a dissection microscope with the mucosa side facing up. After 2 hours of fixation with 4% PFA, tissue was

washed for 3x10 min with PBS. The mucosa-submucosa layer was then separated from the underlying muscularis using fine dissecting forceps. To collect the submucosal plexus, any attached mucosa was gently scraped off using blunt forceps. To collect the myenteric plexus, circular muscles were carefully peeled off from the remaining muscularis using fine dissecting forceps. This revealed the longitudinal muscle-myenteric plexus (LMMP) preparation for further staining.

The submucosal and LMMP preparations were incubated in 1% Triton X-100 in PBS containing 1% bovine serum albumin to permeabilise cell membranes. After 3x10 minutes wash with PBS, tissues were incubated in primary antibodies overnight at 4˚C. Tissues were then washed 3 times for 10 minutes with PBS and incubated in secondary antibodies for 2.5 hours on a shaker at room temperature. Secondary antibodies were washed off with PBS for 3x10 minutes. Depending on the markers of interest, nuclei were stained with DAPI (Sigma-Aldrich, D9542) for 20 minutes at 1:2000 dilution. Tissues were then mounted with DAKO fluorescence mounting medium (Agilent, S3023).

## Muscle cross section staining

A segment of a jejunum (around 1 cm) was collected from each mouse and placed into 10% natural buffered formalin. Tissues for then sent to Monash Histology Platform for paraffin embedding, sectioning (4 μm cross section) and antigen retrieval (citrate buffer). Slides were stained with a conjugated mouse anti-α-smooth muscle actin antibody in PBS for 2.5 hours. The slides were then washed with PBS 3 x 10 minutes before being mounted with DAKO fluorescence mounting medium.

## Image acquisition and analysis

Images are taken by either Nikon Ti-E fluorescence microscope or Nikon A1r confocal microscope with the NIS-Elements imaging software. The 20x lens was used for Nikon Ti-E microscope, and the 20x multi-immersion (glycerol) lens was used for Nikon A1r microscope. All images were saved as.nd2 format for further analysis using FIJI (Image J).

For villous wholemount, groups of villi that could be viewed along their entire length were chosen for image acquisition. At least 6 images were taken per slide. Images were taken as a Z-stack to capture the whole villus structure. Images were analysed using FIJI and set as hyperstacks with all settings default. Images were then Z-projected for maximum intensity values and viewed as composite images. Villi that were straight with little angle and no overlap with other villi were chosen for analysis. Lengths of the villi were determined using the line tool, and areas of villi were determined using the polygon tool. Length and area were analysed for both the entire villi and the blood-vessel network.

For plexus wholemount, cell numbers were quantified using the CellCounter plugin in FIJI. For myenteric neuronal density, 3 random sections were selected from each animal and were imaged either with 10x objective lens or 20x objective lens as 2x2 sections. Neuronal cell bodies were stained with HuC/D or Hu (Table 1), and only neurons with their entire cell bodies included in the images were counted. Density is calculated as the number of neurons/mm$^2$. Myenteric neurochemical marker expressions were analysed either using the same images for density analysis or additional 2–3 sections were taken if a different antibody panel was used. Each neurochemical marker was counted against HuC/D or Hu staining. At least 1 2x2 image or 2 1x1 images were analysed. All neurons with their entire cell bodies in the images were counted and 180 to 420 cell bodies were counted per animal. For submucosal plexus, 4–5 sections were imaged and analysed for counting. Around 100–240 submucosal neurons were counted with their neurochemical maker staining against HuC/D or Hu staining.

**Table 1.** **Primary and secondary antibodies used.**

| Primary antibody | | | |
|---|---|---|---|
| Target | Host | Dilution | Source |
| aSMA | Mouse | 1:800 | Sigma-Aldrich (C6198) |
| Calbindin | Rabbit | 1:1600 | Swant (CB38) |
| Calretinin | Goat | 1:1000 | Swant (CG1) |
| ChAT | Goat | 1:100 | Sigma-Aldrich (AB144P) |
| HuC/D | Mouse | 1:100 | Thermo Fisher (A-21271) |
| Hu (ANNA1) | Human | 1:5000 | Gift from Dr. V. Lennon |
| nNOS | Sheep | 1:1000 | Gift from Dr. P. Emson (#K205) |
| nNOS | Goat | 1:100 | Abcam (AB1376) |
| PGP9.5 | Mouse | 1:100 | Abcam (ab8189) |
| SiglecF | Rat | 1:100 | BD biosciences (552125) |
| VEGFR2 | Goat | 1:100 | R&D Systems (AF644) |
| VIP | Rabbit | 1:500 | Immunostar (20077) |
| Secondary antibodies | | | |
| Host and Target | Fluorophore | Dilution | Source |
| Donkey anti Mouse | AF488 | 1:500 | Jackson ImmunoResearch (705-545-147) |
| Donkey anti Mouse | AF568 | 1:200 | Thermo Scientific (A10037) |
| Donkey anti Rat | AF488 | 1:500 | Thermo Scientific (A10042) |
| Donkey anti Rabbit | AF568 | 1:500 | Thermo Scientific (A21447) |
| Donkey anti Goat | AF647 | 1:500 | Thermo Scientific (A21208) |
| Donkey anti Human | AF488 | 1:500 | Jackson ImmunoResearch (709-545–149) |

For muscle thickness analyses, 4 images were taken per mouse. Circular muscle, longitudinal muscle and total muscularis length were determined using line tool in Fiji. 10 markers were drawn per image per muscle location, and muscle thicknesses were calculated as averages of 40 measurements for each animal.

## Video imaging for intestinal motility

We used the video imaging apparatus described in Swaminathan, Hill-Yardin [33] and adapted the protocol from Neal, Parry [32]. In brief, a 5–7 cm segment of jejunum was first flushed with Krebs-Henseleit (Krebs) solution and placed into a horizontal organ bath with each end cannulated into plastic tubes that are connected to syringes. The hydrostatic pressure was set by the difference between the height of the jejunum and the level of Krebs in the syringes. This pressure (5.5 to 7.5 cm of Krebs) was maintained throughout each experiment. Tissues in the chambers were continuously perfused with warmed Krebs solution to maintain a temperature of 34–37°C and bubbled with carbogen gas (95% $O_2$, 5% $CO_2$). As previously described [32], we noticed jejunum contractility from BALB/c mice decreased approximately 2 hours after the tissue was cannulated. Thus, the recording protocol was designed including 30 minutes of equilibration, 2 x 15 minutes of control recordings, 1 x 15 minutes of recording with DMPP (1,1-Dimethyl-4-phenylpiperazinium iodide, Sigma-Aldrich D5891) application and 1 x 15 minutes of washout recording. 1 µM of DMPP was dissolved in Krebs solution and superfused to the chambers during drug recording. A washout recording was set at the end of each experiment to confirm tissue viability and it is not included for analysis. Videos were recorded in AVI format at 30 frames per second using a webcam (Jiffy c1000). Videos were processed using Scribble 2 and them imported to Analyse 2, a plug-in for MATLAB.

Spatiotemporal heatmaps were generated, and neurogenic contractile patterns were analysed using different functions in Analyse 2 (S4 Fig).

### Real-time polymerase chain reaction

Each segment of a jejunum (1 cm) was collected in 1 mL of Trizol reagent (Invitrogen, 15596018), and the total RNA was isolated following the manufacture's instruction. RNA (1 μg) was reverse transcribed to make complementary DNA using SuperScript VILO cDNA Synthesis Kit (Invitrogen, 11754) using random primers. Real-time PCR was run on the QuantStudio 6 flex instrument. Real-time PCR was performed with 10 uL volume per sample containing 100 ng cDNA using Power SYBR Green PCR Master Mix (Applied Biosystems A25741). 40 amplification cycles were performed according to manufacturer's protocol. Real-time PCRs were performed for IL-4 (forward: TCGGCATTTTGAACGAGGTC, reverse: CAAGCATGGAGTTTTCCCATG), IL-13 (forward: CCTGGCTCTTGCTTGCCTT, reverse: GGTCTTGTGTGATGTTGCTCA), IFNγ (forward: TCCTCATGGCTGTTTCTG, reverse: TCTTCCACATCTATGCCACTTG), and GAPDH (glyceraldehyde 3-phosphate dehydrogenase, forward: CGTCTTCACCACCATGGAGA, reverse: CGGCCATCACGCCACAGTTT). Results were analysed using the Pfaffl method. RNA expression for each animal was determined as a fold change relative to the average expression of the house keeping gene GAPDH from BALB/c control mice.

### Intestinal smooth muscle contractility using organ bath

Intestinal smooth muscle contractility was analysed using an organ bath apparatus as previously described [5]. In brief, 1 cm of jejunum and ileum were hooked between pre-calibrated force transducers (MLT0420) with a tension of 1 gram in the organ bath (Panlab Harvard apparatus). Force transducers were connected to an octal bridge amp and 'Powerlab" data acquisition hardware which allowed the handling of data using Labchart V8 software (ADInstruments). Each chamber of the organ bath was filled with Krebs solution at 37.5˚C and bubbled with carbogen gas (95% $O_2$, 5% $CO_2$). Tissues were equilibrated over 1 hour with 2 buffer changes at 20-minute intervals to maintain viability. After equilibration, acetylcholine was added at 1-minute intervals with progressively increasing doses from $10^{-10}$ M to $10^{-4}$ M. All data were carried out using Labchart V8 (ADInstruments). For calculating the average contraction strength and frequency, a 10-minute timespan was selected where there was no evidence of the tension being externally altered after the final buffer change and before acetylcholine addition. Tissues that were not contracting or did not exhibit a regular cyclic contraction pattern were excluded from analyses.

### Statistics and data presentation

Data were analysed using GraphPad Prism 9 (GraphPad Software). Datasets containing 2 independent groups (genotype and infection status) were analysed with a two-way analysis of variance (ANOVA) followed by Tukey's multiple comparisons post-hoc test. Significant differences for each individual variable and the naïve group are indicated inside each graph. Significance of the overall effect of independent group (genotype or infection status) is indicated under each graph. Data containing two parametric variables were analysed using paired or unpaired student T tests, and data contain two nonparametric variables were analysed using Mann-Whitney test or Wilcoxon sign-ranked test. Data containing more than two parametric variables were analysed using a one-way ANOVA followed by Tukey's multiple comparisons post-hoc test. Significance levels of $p < 0.05$ (*), $p < 0.01$ (**), $p < 0.001$ (***) and

p<0.0001 (****) were chosen. Error bars were shown as mean ± standard deviations for parametric variables.

## Supporting information

**S1 Fig. Nb infection does not influence myenteric ganglion connectivity.** dblGATA1 mice and wildtype BALB/c controls were infected with 400–500 L3 Nb and sacrificed day 9 p.i. Control groups of naïve mice were included. Myenteric plexus layers were collected as detailed in Materials and Methods and stained with antibodies against the pan neuronal marker HuC/D (green) and calbindin (red). Representative images are shown from a single animal from each group. Intra-ganglionic connections were indicated as calbindin[+] axonal bundles between ganglions.
(TIF)

**S2 Fig. Eosinophils are present in the submucosal, but not myenteric, plexus of BALB/c mice.** dblGATA1 mice and wildtype BALB/c controls were infected with 400–500 L3 Nb and sacrificed day 9 p.i. Control groups of naïve mice were included. Myenteric and submucosal plexus layers were collected as detailed in Materials and Methods and stained with antibodies against the pan neuronal marker PGP9.5 (red) and SiglecF (green). Representative images are shown for (A&B) the myenteric plexus and (C&D) the submucosal plexus of naïve and Nb infected BALB/c mice. Inserts in C&D show magnified areas were SiglecF[+] eosinophils can be found in close proximity (<1 μm) with PGP9.5[+] submucosal neurons. (E) Villous whole-mounts from naïve BALB/c mice stained with antibodies against SiglecF (green), αSMA (red) and VEGFR2 (magenta). (F) Representative images of the submucosal plexus from a naive dblGATA1 mouse stained with antibodies against SiglecF (green) and PGP9.5 (red).
(TIF)

**S3 Fig. Spatiotemporal heatmap annotations for analysing neurogenic contractions.** Results from video imaging experiments were analysed using Analyse 2 in MATLAB. The x-axis in the heatmap indicates time and y-axis indicates position of the tissue. Each colour coded pixel indicates the diameter of the jejunum at a given time and location. Neurogenic contractions are indicated by the warm coloured bands in the heatmap. To generate contraction amplitude and slow wave data, a horizontal line was drawn at the distal ¼ of the tissue using the Heatmap Annotation function (A). The software calculates the changes in tissue diameter (indicated by colour) at any time at the line location. Contraction amplitudes (A, blue box) were recorded as % changes in tissue diameter, which is calculated by dividing changes in diameter between relaxation (point x) and contraction (point y) by relaxation, or (x–y)/x. Slow wave information (A, red box) was performed using fast Fourier transformation that separates the high frequency slow wave contraction (red circle and arrow) from the low frequency neurogenic contractions. Contraction velocity and duration were analysed with the Contraction Wave Annotation function (B, zoomed in view of a contraction wave). Contraction duration was defined as how long a PCC contraction lasts within the distal ¼ of the tissue, as indicated by Label 1. Contraction velocity is defined as the absolute value of speed of a PCC contraction that travels from the distal end to the proximal end of the tissue (Label 2).
(TIF)

**S4 Fig. The effects of Nb infection and DMPP application on jejunal neurogenic contractions.** BALB/c mice were infected with 400–500 L3 Nb and sacrificed day 6–9 p.i. Control groups of naïve mice were included. Jejunal contractile activities were recorded under physiological conditions in a horizontal organ bath using a video camera, and data analysed with in-house software Scribble 2.0 and Analyse 2. Analysis of the spatiotemporal heatmap provided

information on PCC contraction duration and velocity, contraction amplitude, and slow wave frequencies. Comparison of (A) neurogenic contraction initiations, (B) peristaltic contraction complex (PCC) frequencies, (C) neurogenic contraction duration and (D) slow wave contraction frequencies recorded in tissues from naïve or infected mice under baseline conditions or after DMPP application. Open and closed symbols indicate recordings taken during baseline conditions (open) or after DMPP application (closed). Symbols represent individual animals and individual experiments (n = 7–8 per group) with each experiment including tissues from one naive animal and one infected animal kept within the individual chambers of the same organ bath. Tissues that cease contracting before the end of each experiment were excluded from analyses. Data are shown as mean ± SEM. For (A&B) significance was determined by Mann-Whitney test when comparing naïve and infected mice and Wilcoxon signed-rank test when comparing baseline and DMPP. For (C&D), significance was determined using either unpaired (for naïve vs infected) or paired (baseline vs DMPP) student T-test.
(TIF)

**S5 Fig. Nb infection increases type 2 cytokine gene expression within the jejunum.**
dblGATA1 mice and wildtype BALB/c controls were infected with 400–500 L3 Nb and sacrificed at day 9 p.i. Control groups of naïve mice were included. The expression of (A) IL4, (B) IL13 and (C) IFNγ were determined by qRT-PCR for whole jejunum tissue lysate. mRNA expressions in each animal were calculated using the Pfaffl method and presented as fold changes to the average of the housekeeping gene GAPDH. Symbols represent individual animals and data are shown as mean ± SEM from one experiment with n = 4 per group. The significance was calculated by two-way ANOVA with Tukey's post-hoc analysis.
(TIF)

**S6 Fig. Nb infection leads to ileal longitudinal muscle hypercontractility.** dblGATA1 mice and wildtype BALB/c controls were infected with 400–500 L3 Nb and sacrificed at the indicated days p.i. Control groups of naïve mice were included. Ileal longitudinal muscle segments were mounted longitudinally within an organ bath as described in the Materials and Methods and (A) contraction frequency and (B) contraction strength recorded under baseline conditions. Thereafter the same tissues were subject to stimulation using incremental doses of acetylcholine and (C) the dose-dependent change in contraction strength or (D) the maximum increase in contraction strength determined. Symbols represent individual animals pooled from 2 independent experiments (n = 8–15 animals/group). Data are shown mean ± SEM and significance determined using a two-way ANOVA with Tukey's post-hoc analysis.
(TIF)

**S7 Fig. Hp infection alters neuron numbers and some aspects of neurochemistry.** BALB/c mice were infected with 200 L3 Hp at day 0 for the primary infection group or day 0 and day 37 for the secondary infection group and were sacrificed day 14 p.i. or day 51 p.i., respectively. Control groups of naïve mice were included. (A) Representative images of myenteric plexus stained with Hu from a single naïve, primary Hp infected, and secondary Hp infected BALB/c mice. (B) Myenteric neuronal cell density was calculated by counting cell bodies and presented as number/mm$^2$. (C&E) Representative images are shown from a single naïve BALB /c animal for myenteric plexus stained with antibodies directed against C) Hu (red), CalB (green), and CalR (cyan) or E) Hu (red), CalB (green), and nNOS (cyan). (D &F) The percentage of total neurons expressing the indicated neurochemical markers was calculated by counting at least 180 Hu$^+$ cell bodies. Symbols represent individual animals with data pooled from 2 separate experiments with n = 8 animals per group. All data are shown as mean ± SEM and significance

determined using a one-way ANOVA with Tukey's post-hoc analysis.
(TIF)

## Acknowledgments

We thank Dr Irena Carmichael at Monash Micro Imaging platform for their support with microscopy.

## Author Contributions

**Conceptualization:** Haozhe Wang, Jaime P. P. Foong, Joel C. Bornstein, Nicola L. Harris.

**Formal analysis:** Haozhe Wang, Kristian Barry, Gillian Coakley, Mati Moyat.

**Funding acquisition:** Nicola L. Harris.

**Investigation:** Haozhe Wang, Kristian Barry, Aidil Zaini, Gillian Coakley, Mati Moyat, Carmel P. Daunt, Lakshanie C. Wickramasinghe, Rossana Azzoni, Roxanne Chatzis.

**Methodology:** Haozhe Wang, Kristian Barry, Gillian Coakley, Mati Moyat.

**Project administration:** Nicola L. Harris.

**Resources:** Bibek Yumnam, Mali Camberis, Graham Le Gros, Benjamin J. Marsland, Nicola L. Harris.

**Supervision:** Olaf Perdijk, Jaime P. P. Foong, Joel C. Bornstein, Benjamin J. Marsland, Nicola L. Harris.

**Validation:** Haozhe Wang, Kristian Barry, Aidil Zaini, Gillian Coakley, Mati Moyat, Carmel P. Daunt, Lakshanie C. Wickramasinghe.

**Visualization:** Haozhe Wang, Kristian Barry, Gillian Coakley, Mati Moyat.

**Writing – original draft:** Haozhe Wang, Nicola L. Harris.

**Writing – review & editing:** Haozhe Wang, Olaf Perdijk, Nicola L. Harris.

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
