## [Decision Letter · Decision Letter 0]

27 Nov 2023

Dear Prof Harris,

Thank you very much for submitting your manuscript "Helminth infection driven gastrointestinal hypermotility is independent of eosinophils and mediated by alterations in smooth muscle instead of enteric neurons" for consideration at PLOS Pathogens. As with all papers reviewed by the journal, your manuscript was reviewed by members of the editorial board and by several independent reviewers. In light of the reviews (below this email), we would like to invite the resubmission of a significantly-revised version that takes into account the reviewers' comments.

Your manuscript has been reviewed by three expert reviewers, all of whom appreciated the importance of your study. However, reviewers 1 and 2 have raised concerns regarding the relevance of current study to other helminth models with chronic infections. In light of these comments, I would like to invite you to revise your manuscript to address the points raised by the reviewers. The detailed evaluation reports are attached to this letter for your reference.

We cannot make any decision about publication until we have seen the revised manuscript and your response to the reviewers' comments. Your revised manuscript is also likely to be sent to reviewers for further evaluation.

Sincerely,

Shahid Siddique

Guest Editor

PLOS Pathogens

Margaret Phillips

Section Editor

PLOS Pathogens

Kasturi Haldar

Editor-in-Chief

PLOS Pathogens

orcid.org/0000-0001-5065-158X

Michael Malim

Editor-in-Chief

PLOS Pathogens

orcid.org/0000-0002-7699-2064

Thank you for submitting your manuscript to PLOS Pathogens. It has been reviewed by three expert reviewers, all of whom appreciated the importance of your study. However, reviewers 1 and 2 have raised concerns regarding the relevance of current study to other helminth models with chronic infections. In light of these comments, I would like to invite you to revise your manuscript to address the points raised by the reviewers. The detailed evaluation reports are attached to this letter for your reference.

Reviewer's Responses to Questions

**Part I - Summary**

Reviewer #1: Wang et al present a manuscript investigating the physiological mechanisms by which worm infections stimulate intestinal motility. Previous research involving research with other pathogens has highlighted varying mechanisms of how the gut responds to infection, and the current paper tries to address how this occurs during acute infection with a GI parasite, Nippo. Their hypothesis was that Nippo infection would alter enteric neuron function, perhaps via eosinophils, but neither of these hypothesis were true, instead the infection seems to directly act on smooth muscle.

The authors also use a range of complementary techniques, including some nice imaging and measures of contraction.

Overall it is a well conducted study and the data are well presented, although the advances in knowledge of host-parasite interactions are fairly incremental. The paper is also quite descriptive, and a lot of "negative" data are shown, particularly figures regarding independence of eosinophils for these effects seen. A few comment and questions are listed below:

Reviewer #2: The manuscript by Wang et al. shows that intestinal helminth infection is associated with substantial changes in smooth muscle thickness and tensile strength but surprisingly, little change in the local enteric nervous system (ENS), which is thought to play a key role in regulating intestinal motility. The changes in smooth muscle physiology were not dependent upon the presence of eosinophils. This is an interesting and nicely presented study, unpacking many of the assumptions that we have regarding the regulation of worm expulsion during Nb infection. While many of the data pieces presented are negative, these results are important in identifying the important changes that occur in intestinal physiology during helminth infection. As such, these data represent an important contribution to the literature that will inform future research efforts.

Reviewer #3: The paper by Want et al. is focused on characterizing intestinal motility in Nippostrongylus brasiliensis infection and the role of eosinophils and the enteric nervous system. They investigate several aspects of physiological alterations of the intestine and the effect that eosinophils play. In their observations of neuronal density I appreciated their caveat that the reduction of mysenteric neurons could result from tissue growth rather than loss of neurons. Part of their results suggested that neither Nb infection nor eosinophils had an impact on intestinal ENS structure. I was also interested to see that no eosinophils could be found in the myenteric plexus in naïve or infected mice, nor could they be found in the muscle wall.

Overall the paper is well written, the experimental design and progression is logical and clearly conveyed. This study represents a significant advancement to the field and will be of interest to a broad readership. I had only a few minor suggestions. Well done!

**Part II – Major Issues: Key Experiments Required for Acceptance**

Reviewer #1: 1. It is difficult to know how representative these results are for helminth infections, ie are they specific to acute Nippo infections, or would similar results be seen in any other helminth models? It would be interesting to see if similar mechanisms operate in models such as secondary H. poly infection-

2. The main positive finding of the study is that Nippo infection increases smooth muscle contractillity, but that has been well reported in the literature and the cytokines that drive this response have been well characterised (IL-4/13), which the authors acknowledge. Can the authors elaborate on what information the present study has added in this area? Or, is the main area of novelty the fact that neurons and eosinophils are NOT involved much?

Reviewer #2: Nippostrongylus brasiliensis is expelled quickly from the murine intestine, whereas most endemic parasitic helminths chronically reside in the small intestine in infected humans and other mammals. While Nippo infection is an important model used to study the intestinal physiology associated with worm expulsion, expulsion is not the norm. How does chronic infection with Heligmosomoides polygyrus influence intestinal ENS responses and smooth muscle function? It seems beyond the scope of the current report to investigate the role of eosinophils in these responses in H. polygyrus infection, but investigation of how acute vs. chronic infection affects smooth muscle responses to infection is important in establishing the physiological relevance of the data presented.

Reviewer #3: (No Response)

**Part III – Minor Issues: Editorial and Data Presentation Modifications**

Reviewer #1: 3. Lines 123: I would consider adding data from Fig S1 to the main figure set, particularly because the authors have admitted it was "interesting"

Reviewer #2: 1. The coding for the days and genotypes is quite complex, as the reader needs to apply the single “legend” across all 4 panels, and the 4 panels have data from different days post-infection. Is there a way to simplify?

2. The stats indications on the bottom of the panels (genotype vs. infection) are confusing in conjunction with the stats already indicated on the graphs.

Reviewer #3: Much of the paper is about the physiology and physiological changes in the enteric nervous system (ENS) and the myenteric and submucosal plexus. This is introduced in lines 56-68 but it comes back later in the results and discussion. I think a diagram or illustration of these tissues and their relation to lamina propria etc would be informative for the reader and would facilitate understanding of the data. I suggest that the authors add this as a new figure, or perhaps as a panel on existing figure 1.

Minor suggestions:

Line 169: It’s a bit confusing as currently written. I suggest changing to “Ahrends and Aydin et al., reported that…

PLOS authors have the option to publish the peer review history of their article (what does this mean?). If published, this will include your full peer review and any attached files.

Reviewer #1: No

Reviewer #2: No

Reviewer #3: No
---

## [Editor Report · Decision Letter 1]

29 Jul 2024

Dear Prof Harris,

We are pleased to inform you that your manuscript 'Helminth infection driven gastrointestinal hypermotility is independent of eosinophils and mediated by alterations in smooth muscle instead of enteric neurons' has been provisionally accepted for publication in PLOS Pathogens.

Best regards,

Shahid Siddique

Guest Editor

PLOS Pathogens

Margaret Phillips

Section Editor

PLOS Pathogens

Michael Malim

Editor-in-Chief

PLOS Pathogens

orcid.org/0000-0002-7699-2064
---

## [Editor Report · Acceptance letter]

8 Aug 2024

Dear Prof Harris,

We are delighted to inform you that your manuscript, "Helminth infection driven gastrointestinal hypermotility is independent of eosinophils and mediated by alterations in smooth muscle instead of enteric neurons," has been formally accepted for publication in PLOS Pathogens.

Best regards,

Michael Malim

Editor-in-Chief

PLOS Pathogens

orcid.org/0000-0002-7699-2064